# Exploring the Exposome Spectrum: Unveiling Endogenous and Exogenous Factors in Non-Communicable Chronic Diseases

**DOI:** 10.3390/diseases12080176

**Published:** 2024-08-02

**Authors:** Laura Di Renzo, Paola Gualtieri, Giulia Frank, Rossella Cianci, Mario Caldarelli, Giulia Leggeri, Glauco Raffaelli, Erica Pizzocaro, Michela Cirillo, Antonino De Lorenzo

**Affiliations:** 1Section of Clinical Nutrition and Nutrigenomics, Department of Biomedicine and Prevention, University of Rome Tor Vergata, Via Montpellier 1, 00133 Rome, Italy; laura.di.renzo@uniroma2.it (L.D.R.); giulia.leggeri@uniroma2.it (G.L.); delorenzo@uniroma2.it (A.D.L.); 2PhD School of Applied Medical-Surgical Sciences, University of Rome Tor Vergata, Via Montpellier 1, 00133 Rome, Italy; glauco.raffaelli@yahoo.it (G.R.); ericapizzocaro@gmail.com (E.P.); 3School of Specialization in Food Science, University of Tor Vergata, Via Montpellier 1, 00133 Rome, Italy; michela.cirillo@unifi.it; 4Department of Translational Medicine and Surgery, Catholic University of the Sacred Heart, 00168 Rome, Italy; mario.caldarelli01@icatt.it; 5Fondazione Policlinico Universitario A. Gemelli, Istituto di Ricovero e Cura a Carattere Scientifico (IRCCS), 00168 Rome, Italy

**Keywords:** exposome, non-communicable chronic diseases, diet

## Abstract

The exposome encompasses all endogenous and exogenous exposure individuals encounter throughout their lives, including biological, chemical, physical, psychological, relational, and socioeconomic factors. It examines the duration and intensity of these types of exposure and their complex interactions over time. This interdisciplinary approach involves various scientific disciplines, particularly toxicology, to understand the long-term effects of toxic exposure on health. Factors like air pollution, racial background, and socioeconomic status significantly contribute to diseases such as metabolic, cardiovascular, neurodegenerative diseases, infertility, and cancer. Advanced analytical methods measure contaminants in biofluids, food, air, water, and soil, but often overlook the cumulative risk of multiple chemicals. An exposome analysis necessitates sophisticated tools and methodologies to understand health interactions and integrate findings into precision medicine for better disease diagnosis and treatment. Chronic exposure to environmental and biological stimuli can lead to persistent low-grade inflammation, which is a key factor in chronic non-communicable diseases (NCDs), such as obesity, cardiometabolic disorders, cancer, respiratory diseases, autoimmune conditions, and depression. These NCDs are influenced by smoking, unhealthy diets, physical inactivity, and alcohol abuse, all shaped by genetic, environmental, and social factors. Dietary patterns, especially ultra-processed foods, can exacerbate inflammation and alter gut microbiota. This study investigates the exposome’s role in the prevention, development, and progression of NCDs, focusing on endogenous and exogenous factors.

## 1. Introduction

The exposome is an evolving interdisciplinary concept that refers to the full range of endogenous and exogenous exposure an individual encounters during a lifetime. The sources of exposure are many and include biological, chemical, and physical agents, as well as a wide plethora of environmental agents, such as psychological, relational, and socioeconomic factors [1]. Exposure assessment also considers the duration and intensity of contact with these sources, as they play a key role in characterizing the overall interaction with the environment [2].

Beyond identifying exposure sources, the focus is also on the complex interplay between stressors and their relationship to human health over time. This multidisciplinary approach involves a wide range of scientific disciplines, including toxicology, that plays a crucial role in understanding the mechanisms of the long-term effects of toxic exposure [3]. Indeed, it is well documented how different factors, such as air pollution, racial background, and socioeconomic status, can contribute to the onset of several diseases [4]. For example, it has been shown that advanced glycation end products (AGEs) [5], toxic advanced glycation end products (TAGEs) [6] and polycyclic aromatic hydrocarbons (PAHs) [7] play a crucial role in the onset of several diseases, such as metabolic, cardiovascular, neurodegenerative, kidney, liver disease, infertility, and cancer [8].

An exposome analysis requires developing and implementing sophisticated tools and methodologies to assess the wide range of interactions that affect human health while attempting to minimize confounding effects. For example, accurate analytical methods have been developed to measure contaminants in biofluids, food, air, water, and soil [9].

They are often limited to specific chemical agents, which may underestimate the real risk, as the number of chemical agents continues to increase. The complex interactions between chemical agents and environmental factors influence biological responses, underlining the importance of integrating exposomics into precision medicine to optimize the diagnosis and treatment of several diseases [10].

Prolonged exposure to environmental and biological stimuli that trigger the inflammatory response can induce a state of chronic inflammation. This leads to the prolonged activity of immune cells, including lymphocytes, macrophages, and plasma cells, within the tissue, along with the sustained release of pro-inflammatory cytokines, chemokines, and other molecules [11]. This persistent chronic low-grade inflammation plays a significant role in the initiation and progression of several chronic non-communicable diseases (NCDs) [12]. NCDs represent a crucial node in the public health context, and include obesity, cardiometabolic disorders, cancer, respiratory diseases, autoimmune conditions, and depression [13]. NCDs are linked to smoking, unhealthy diet, physical inactivity, and alcohol abuse. These factors, in turn, are influenced by genetic, environmental, and social factors [14]. Dietary factors significantly promote inflammation, contributing to the increasing prevalence of NCDs. Specific foods, such as ultra-processed ones, often contain additives and chemicals that can exacerbate inflammation, as well as potentially harmful substances from processing or packaging methods [11]. Moreover, it is well known that dietary patterns could influence the gut microbiota composition [15], encouraging the growth of specific bacterial phyla [16].

This review of reviews aims to investigate the role of the exposome in NCD prevention, development, and progression, focusing on the specific responsibilities of the endogenous and exogenous factors.

## 2. Materials and Methods

The literature review was conducted in March 2024 using scientific databases, including PubMed, Google Scholar, and Cochrane Library. The focus was on identifying reviews related to non-communicable diseases and the exposome. The filters used were “free full text”, “Review” and “5 years”.

Specific keywords were employed to refine the search: “non-communicable chronic disease and endogenous factor”, “non-communicable chronic disease and exogenous factor”, “non-communicable chronic disease and advanced glycation end products”, “non-communicable chronic disease and toxic advanced glycation end products”, “non-communicable chronic disease and polycyclic aromatic hydrocarbons”, “non-communicable chronic disease and acrylamide”, and “non-communicable chronic disease and exposome”. Only reviews published in peer-reviewed journals that explicitly mentioned the exposome and disease in their title, abstract, or text were considered for inclusion.

The search process was carried out independently by six operators following the Preferred Reporting Items for Systematic Reviews and Meta-Analyses (PRISMA) guidelines and the REAPPRAISED checklist. The selection procedure began with an initial examination of the titles, followed by the abstracts, and subsequently the full texts. Duplicate articles were identified and removed after a thorough review of the titles. In total, 61 articles were found using the keywords, 2 articles were excluded as they were duplicates, 9 articles were excluded after reading the abstracts, 8 articles were excluded after full reading, and 1 article was excluded as it was not in English. Ultimately, 41 articles meeting the criteria and relevant to the topic were included, comprising 41 reviews.

The flow chart inherent to the choice of studies is illustrated in Figure 1.

## 3. Endogenous Factors

### 3.1. Exposome, Genetics, and Epigenetics

Considering that the exposome encompasses a series of environmental factors that can be modified, it is possible to improve habits to lead a healthier life and be less exposed to substances highly harmful to organisms. The influence of genetics on health has been considered for decades, but recently a new scientific concept has been coined, known as the exposome, which is described as the set of environmental sources that have a huge impact on our body from conception to the end of life.

Environmental chemicals, air, land, water, and food pollutants are largely distributed in the environment and can exert, at various levels, tissue damage, with the release of damage-associated molecular pattern molecules (DAMPs), activation of innate immunity receptors, such as Toll-like receptors (TLRs), and consequent activation of adaptive immunity. Furthermore, the development of neoantigens due to the action of environmental chemicals can prompt pro-inflammatory cytokines production, resulting in a reduction in T-regulatory cells (Tregs) and activation of autoreactive T- and B-cells, leading to the progress of autoimmunity [17]. Environmental toxicants elicit oxidative stress, generating reactive oxygen species (ROS) production and directly damaging DNA; moreover, they alter DNA methylation and consequent gene expression. Thus, the combination of genetic and environmental factors can lead to the different phenotypes of autoimmune diseases. As in the case of allergic diseases, the hygiene hypothesis has been proposed also in autoimmune disorders: changes in lifestyle and dietary habits and the use/abuse of antibiotics in foods can reduce microbiota diversity and consequently dysregulate the activity and function of the related immune system and activate of pro-inflammatory pathways [18].

The role of the exposome in association with genetic predisposition is being studied in the case of cancer. Even if the mechanisms are not completely understood, the exposome can affect epigenetic mechanisms involved in histone methylation that regulates gene expression [19]. The promoter methylation can downregulate the interaction with transcription factors and RNA-polymerase and, consequently, with gene transcription and protein production. Some micronutrients, such as folic acid, and group B vitamins act as substrates for the correct function of gene expression. It is important to highlight the epigenetic impact of the environment during pregnancy, delivery, maternal lifestyle, infant feeding and the long-term effect of an unhealthy diet on the fetus, microbiota development, and shaping.

Environmental pollutants can drive inflammatory response via micro-RNAs, through different genes and DNA modifications. After DNA damage, interleukin (IL)-1β and tumor necrosis factor (TNF)-α are activated [20]. Another mechanism involved in the onset of neoplastic diseases is oxidative stress, which can reduce the anti-inflammatory response sustained by Tregs and activate a chronic low-grade inflammation that can lead to neoplastic transformation.

### 3.2. Exposome and Microbiota

During the last decades, microbiota have been recognized to play a pivotal role in the onset and progression of several NCDs. The large number of bacteria, fungi, viruses, archaea, and other microorganisms colonizing several epithelia of our body is needed for shaping and regulating the immune system [21]. Environmental factors, such as several pollutants and the consequences of climate change, have shown strong modifications in microbiota composition, with an increased number of pathogens and a reduction in commensal bacteria able to compete for colonization. This situation, named dysbiosis, leads to the alteration of gut permeability and the onset of inflammation and disease [22]. The gut microbiota (GM) are different in different stages of life from birth, when bacterial colonization occurs, to adulthood, when it is in equilibrium with the host, to senescence, when it is related to physical conditions. At several sites of the body, microbiota interact with the immune system, shaping immune cell maturation and function, and with xenobiotics, such as antibiotics, drugs, hormones, and environmental factors, such as diet, food supplements, toxins, pollutants. The redundancy of microbiota is linked to the capability to respond to life stressors to preserve the host equilibrium [23]. Therefore, when the microbial diversity or richness index are reduced, the inflammation occurs.

The role of the exposome is well explained in case of allergy. The development of allergic diseases is linked to the intricate interplay between genetics and environmental factors. While genetic factors are well established due to several known polymorphisms related to the susceptibility to allergy, environmental factors are still a matter of discussion [24]. The ‘hygiene hypothesis’, modifications in food consumption, increase in pollutants, climate change, and the loss of biodiversity play a key role in influencing the onset of allergic diseases. The epithelial barrier theory represents a growing matter of evidence, suggesting that exposure to environmental factors can disrupt the epithelial barriers at different levels (skin, oral, gut, lung, and genitourinary tract) and a leaky barrier can permit the transfer of pathogens in a dysbiotic state and the activation of inflammatory pathways [25].

Intestinal barrier disruption is associated with allergic disorders and autoimmune diseases, such as rheumatoid arthritis, multiple sclerosis, and type 1 diabetes [26].

Several bacteria and microorganisms have been related to the onset and progression of neoplastic lesions, such as *Helicobacter pylori*, which is considered a human class I carcinogen for gastric cancer, and MALT-lymphoma, and other oncogenic pathogenic microorganisms, such as *Escherichia coli*, *Streptococcus bovis*, *Bacteroides fragilis*, *Fusobacterium nucleatum*, and some Enterobacteriaceae. They are linked to colorectal cancer through their capability to damage, the epithelial and mucosal layer, cells, and DNA and elicit a pro-inflammatory and carcinogenetic immune response [27].

Moreover, another mechanism of action of microbiota is represented by the capability of GM metabolites to interact with epigenomic mechanisms [28]. Dietary fibers undergo saccharolytic fermentation with subsequent production of short/chain fatty acids (SCFAs), such as butyrate, propionate, and acetate. SCFAs reduce the expression of pro-carcinogens [21]. Furthermore, SCFAs bind secondary bile acids, which are other GM metabolites, able to exert oxidative DNA damage and tumorigenesis. The role of butyrate in inhibiting histone deacetylase and subsequent oncogenic pathways is controversial, as well as its role in promoting T-regulatory cells; however, it is known that it reduces the pro-oncogenic miRNA in colonic cancer, downregulates pro-inflammatory cytokines, such as IL6, and enhances the production of anti-inflammatory IL-10. Moreover, other microbial metabolites can activate the nuclear aryl hydrocarbon receptor (AhR), which is able to modify the expression of genes related to inflammation, and it is related to immunotolerance. On the other hand, the activation of AhR is linked to mechanisms of tumor escape. Other GM metabolites, derived from metabolism of phosphatidylcholine, choline, and carnitine from red meat, and milk), such as trimethylamine N-oxide (TMAO), can promote NOD, LRR, and pyrin domain-containing protein 3 (NLRP3) inflammasome activation and pro-inflammatory cytokines, such as TNF and IL-1β, and are linked to tumor and metastasis progression, probably through chronic inflammation, oxidative stress, and DNA damage [29].

Another group of GM metabolites is represented by secondary bile acids, mainly metabolized by *Bacteroides fragilis*, *Clostridium perfringens*, *Lactobacillus*, and *Bifidobacterium*. They prompt the malignant transformation of colonic adenomas.

The alteration in the intestinal microbiota composition can impact bile acid metabolism. Therefore, microbiota itself can contribute to shape the same microbiome community in terms of different functions, stability, and diversity. The results of these changes are metabolites that themselves can be detrimental for the GM. The gastrointestinal exposome include a wide range of gut exogenous and endogenous factors [23]. The exogenous exposome is derived from nutrients and other xenobiotics and can modify the GM equilibrium; the endogenous exposome is derived from the intricate relationship among cytokines, hormones, and other mediators of immune system. In this way, the GM composition results in a dynamic equilibrium state; the disruption of this equilibrium prompts the mucosal and systemic diseases.

### 3.3. Aging

Worldwide, an increasing number of people aged 60 and above are altering the demographics and leading to a rise in age-related NCDs with Western countries being particularly affected.

Aging is a biological process characterized by a progressive decline in intrinsic biological functions. This physiological (and metabolic) decline, due to genomic instability, loss of proteostasis, shortening/dysfunction of telomeres, and epigenetic modifications, is identified as the primary risk factor for several human diseases [30].

It is widely recognized that all age-associated diseases share common underlying cellular and molecular mechanisms of aging. These pathways are genetically determined, but environmental and lifestyle factors play a critical role in their regulation.

Indeed, the exposome, broadly defined as the totality of all environmental exposure from conception till death, accelerates the process of aging by affecting the internal biological pathways and signaling mechanisms that result in the worsening of human health.

The heart and vascular system are highly vulnerable to various environmental agents. Tobacco smoking, extreme temperatures, particularly heat, and air, water, and soil pollution are among the most important environmental risk factors for cardiovascular death. The cellular and molecular mechanisms of aging that contribute to the pathogenesis of age-related cardiovascular diseases, lead to a dysfunctional endothelium through elevated production of ROS. Moreover, chronic exposure to pollutants can result in the accumulation of DNA damage, impairing the cell’s ability to repair and preserve genomic integrity. Consequently, this triggers cellular senescence, a state of irreversible growth arrest characterized by significant phenotypic alterations [31]. This senescence-driven inflammatory milieu contributes to tissue dysfunction and promotes accelerated aging, which in turn leads to the activation of immune cells and the production of pro-inflammatory cytokines (i.e., IL-6, IL-1, HMGB1, and S100); chemokines (i.e., IL-8, MCP-1); soluble receptors; metalloproteases; and certain protease inhibitors and growth factors [32].

External exposure include air pollutants, which represent the second risk factor of NCDs. It is recognized that long-term exposure to fine particulate matter 2.5 (PM_2.5_) contributes to 2.9 million global deaths each year, with nearly 50% of these attributable to NCDs such as ischemic heart disease and stroke [31]. More than 90% of the global population breathes air rich in PM which causes damage through the generation of oxidative stress, inflammatory responses, and endothelial dysfunction in the vascular system, transmigrating directly from the lung epithelium into the bloodstream [33]. Air pollution can not only affect the cardiovascular system but can also exert negative effects on our respiratory system and human skin, leading to skin aging and possibly inducing skin cancers [34].

In carcinogenesis, telomere lengths have an important role. Telomeres play as biological clocks that measure the lifespan of a cell and an organism. Particulate air pollution exposure can affect the telomere mitochondrial axis of aging. To demonstrate the chronic health effects of air pollution, the telomere length was proposed as a proxy to assess the exposome. It is not only PM that is negatively correlated with the length of telomeres, but also occupational exposure (polycyclic aromatic hydrocarbons and toxic metals) and tobacco exposure.

Several heavy metals and their derivatives are implicated in accelerating human aging; in particular, lead (Pb) is known to affect the aging brain, causing neurodegenerative diseases, such as Alzheimer’s and Parkinson’s disease, in addition to its ability to impact the epigenome (i.e., DNA methylation and histone modifications) and miRNA expression across different organisms [33].

The possibility of maintaining telomeres in normal cells, through diet and lifestyle interventions, has generated widespread interest as a means of increasing health span and preventing multiple age-related diseases. The traditional Mediterranean diet consists of a high intake of vegetables, fruits, nuts, legumes, and whole grains, a high intake of olive oil, and a low intake of saturated lipids. It is believed that this may exert a multifactorial protective effect reducing disease risk through attenuating specific aging mechanisms (i.e., oxidative stress and inflammation) and possibly influencing telomere length [35]. Therefore, future epidemiological studies and clinical trials should consider nutritional interventions targeting to improve cellular senescence, telomere shortening, and chronic inflammation, all of which are hallmarks of aging. This is especially crucial in obese individuals with comorbidities, as such interventions may influence adipose tissue physiology and mitigate further damage.

### 3.4. Exposome and Obesity

A recent study connected principal component analysis (PCA) and k-means clustering to determine exposure profiles during both the prenatal and postnatal stages, using logistic regression to evaluate the risk of obesity based on the transitions between prenatal and postnatal exposure clusters [36].

Even, indoor air quality has been linked with the development of obesity through the use of markers such as branched chain amino acid (BCAA) and acylcarnitine [37].

The first study to demonstrate this was the HELIX study. The aim of this study was to evaluate the correlation between various environmental factors and childhood obesity, specifically, 77 cases of prenatal exposure and 96 cases of childhood exposure, such as those caused by indoor and outdoor air pollutants, green spaces, tobacco smoking, and chemical pollutants. The results demonstrated a positive correlation between exposure to multiple environmental factors and childhood obesity. Blood levels of copper and cesium were associated with a higher BMI, whereas blood levels of molybdenum and cobalt were associated with a lower BMI [38].

Significant changes in the GM have been described in overweight people. It has been shown that their GM had an increase in *Bacteroides fragilis*, *Fusobacterium*, *Lactobacillus reuteri*, and *Staphylococcus aureus* and lower levels of *Lactobacillus plantarum*, *Methanobrevibacter*, *Akkermansia muciniphila*, *Dysosmobacter welbionis*, and *Bifidobacterium animalis* [39]. Connected to the GM is AHR, a transcription factor that, after binding to a ligand, moves into the nucleus and pairs with ARNT to form an AHR/AHR nuclear translocator (ARNT) complex. This dimer promotes the expression of various target genes such as Ahrr, Cyp1a1, Cyp1b1, and IL22. Additionally, AHR can interact with RelB to stimulate the expression of cytokines and chemokines. There is substantial evidence indicating that AHR plays a role in regulating adaptive immune responses, which are relevant to the development of obesity [40].

For example, supplementing the diet of mice with an AHR ligand can improve metabolic syndrome factors, including hepatic triglyceride levels, fasting glucose levels, and insulin in mice [41].

Moreover, germ-free mice colonized with fecal microbiota from high-fat diet showed reduced AHR activity compared to those colonized with microbiota from conventional diet-fed mice. These findings suggest that a decrease in microbiota-derived AHR ligands is associated with a shift in the composition of gut microbiota in obese humans [40].

In summary, some studies showed that AHR is involved in regulating CYP1B1 expression, which promotes fatty acid synthesis. Therefore, dietary intake of AHR antagonist can prevent fatty liver and obesity in animal models fed with a Western diet [40].

Infectious diseases are another cause of predisposition to obesity. Some studies have shown how viral infections promote adipogenesis. Although the precise mechanisms through which the infection determines adipogenesis aren’t still known, a direct role of viral agents of the adenovirus family has been hypothesized through a direct modification of transcriptional factors and enzymes with an increase in triacylglycerol, ROS, and inflammation.

Another interesting correlation involves circadian rhythms. Sleep disorders, nocturnal exposure to light, and shift work cause alterations in metabolism that predispose one to obesity. Lack of sleep causes an increase in ghrelin levels and a reduction in leptin, resulting in an increased sense of hunger. Furthermore, sleep disorders affect the pituitary–adrenal axis with a growth in cortisol levels related to an increase in abdominal adipose tissue.

The urban environment is also related to the development of obesity. Some studies have shown that a high BMI was correlated with living in low-income neighborhoods and with below-average home economic values.

Moreover, progressive global warming affects obesity by inducing a more sedentary lifestyle and dysregulation of thermogenesis.

Finally, genetic factors also contribute to the development of obesity which shares biological bases with the polygenic form. Genome-wide analyses have not found clear specific genetic loci, suggesting that environmental factors have a greater impact than parental genetics. An interesting observation is the correlation between the age of the mother and the weight of the child: older women are more likely to give birth to children with above-normal weight at birth. Furthermore, given the general increase in BMI, children of overweight couples may have a genetic predisposition even if the biological bases have not been fully clarified [39].

The characteristics of all articles on endogenous factors are listed in Table 1.

## 4. Exogenous Factors

### 4.1. Diet

Epigenetic changes, influenced by environmental factors, alter gene expression without DNA sequence changes. Processes like histone modifications and DNA methylation impact cell cycle stages and disease development. NCDs are linked to epigenetic dysregulation, exacerbated by environmental factors. Indeed, environmental toxicants pose a significant risk to human health by exacerbating chronic age-related diseases, particularly NCDs, such as cardiovascular diseases (CVDs), neurodegenerative diseases, pulmonary diseases, and cancer. These toxins contribute to the development and progression of NCDs through various mechanisms including oxidative stress, inflammation, and mitochondrial dysfunction.

Moreover, exposure to environmental endocrine-disrupting chemicals (EDCs) poses significant risks to human health across various physiological systems, including glucose metabolism, obesity, blood pressure regulation, and reproductive health. Kumar et al. [42] reported that these chemicals disrupt endocrine processes, leading to metabolic imbalances and interfering with adipogenesis, ultimately contributing to obesity and related metabolic disorders, such as type 2 diabetes. Additionally, EDCs impact blood pressure regulation and reproductive health, influencing puberty, fertility, and the occurrence of reproductive disorders. Particularly, Kumar et al. [42] highlighted how nutrition plays a crucial role in mitigating the effects of EDC exposure. Ensuring proper nutrition during critical periods, such as pregnancy, can help mitigate the adverse effects on neurodevelopment and thyroid function. Maintaining a balanced diet rich in nutrients essential for thyroid function, such as iodine and selenium, can support individuals exposed to EDCs. Furthermore, dietary choices, such as consuming fresh, whole foods, and avoiding processed and packaged items, can help minimize exposure to EDCs.

Pandics et al. [31] reported that environmental toxins also impact musculoskeletal health, contributing to conditions, such as osteoporosis, sarcopenia, and arthritis. Poor air quality, lead exposure, and pesticides can worsen these conditions by promoting inflammation, oxidative stress, and mitochondrial dysfunction. To mitigate the detrimental effects of environmental toxins on musculoskeletal health, Pandics et al. [31] emphasized that it is essential to incorporate dietary strategies rich in antioxidants and anti-inflammatory nutrients.

Dietary patterns, including calorie density and macronutrient composition, affect epigenetic modifications, with deficiencies in methyl donors contributing to abnormalities. Particularly, Khajebishak et al. [43] reported that vitamins, such as A, D, E, and K, along with water-soluble vitamins, such as B2, B3, B6, B9, B12, and C, play crucial roles in epigenetic regulation, influencing gene expression and DNA stability. Deficiencies in these vitamins increase the risk of chronic diseases, highlighting the importance of adequate intake for maintaining health and preventing disease.

Indeed, unhealthy diets and insufficient physical activity are significant global health risks, with diet being a key component affecting health and well-being. Suboptimal diet quality, composition, and excessive food intake, combined with a lack of physical activity, often lead to overweight and obesity. Skýbová et al. [14] reported that constant dietary risks were responsible for 11 million deaths globally in 2017, with cardiovascular disease being the leading cause of diet-related death in adults under 70, followed by cancer and type II diabetes. Moreover, the authors highlighted that while many studies have investigated the association between dietary habits and NCDs, few have focused on the relationship between dietary habits and life expectancy without disease. Existing research often relies on alternative indices of healthy nutrition or dietary recommendations based on long questionnaires about dietary frequency, leading to inconsistencies in evidence. Methodological issues, such as geographically unrepresentative data on food consumption, imperfectly described distribution of dietary intake, bias in evaluating diet from various sources, and standardized intake assumptions without considering individual differences further complicate the evaluation of diet’s impact on health outcomes.

Unhealthy diets are marked by contemporary methods of food production, which involve employing elevated temperatures, high pressure, dehydration, decompression, irradiation, salt, and preservatives. These techniques are aimed at prolonging the shelf life and enhancing the taste of food, but they lead to substantial alterations in proteins and lipids. Consequently, post-translational modifications occur, including the formation of advanced AGEs within food items. The AGE formation through the Maillard reaction presents a dual influence on health outcomes. Jansen et al. [44] reported that endogenously formed AGEs contribute to protein dysfunction and inflammation, associated with age-related diseases, such as cataracts and Alzheimer’s disease. Conversely, exogenously formed AGEs, originating from food processing, impact protein digestibility and may have systemic health effects upon consumption. While in vitro studies suggest reduced protein digestibility, conflicting findings in vivo regarding amino acid bioavailability complicate our understanding of it. Human intervention trials linking dietary AGEs to health outcomes yield mixed results, necessitating a critical re-evaluation of study designs. Furthermore, Jansen et al. [44] emphasized that dietary protein-bound AGEs may interact with intestinal epithelial cells, potentially affecting gut barrier integrity and inflammation. Increased AGE consumption correlates with heightened intestinal permeability, dependent on protein glycation degree and inflammatory conditions. In healthy models, AGEs mainly influence tight junction expression without inducing inflammation, while in inflammatory bowel disease models, they may have a protective effect against inflammation. These effects likely involve mechanisms independent of the receptor for AGEs (RAGE), mediated by intracellular signaling pathways. Moreover, Jansen et al. [44] highlighted the effects of dietary protein-bound AGEs on the intestinal epithelium, focusing on pro-inflammatory and potential cancer-promoting properties. While some studies suggest a cancer-promoting effect mediated by RAGE-dependent pathways, methodological limitations, like inadequate controls and lack of dose–response assessment, urge cautious interpretation. Additionally, research on free AGEs presents conflicting results regarding their impact on intestinal epithelial cells, with some suggesting pro-inflammatory effects in vitro but no inflammation observed in healthy or colitis mouse models.

Cruz et al. [45] emphasized the impact of dietary components on 1,2-dicarbonyl compounds and related substances and their role in oxidative stress and glycation processes. Notably, they highlighted the α-dicarbonyl compounds, particularly methylglyoxal and 3-deoxyglucosone, increase during the postprandial state, influenced by factors like food composition and digestion processes. Indeed, they reported the potential of dietary fiber and phenolic compounds to mitigate the levels of these harmful compounds. Fiber, especially those rich in polyphenols, demonstrates scavenging efficacy for reactive carbonyl species, potentially reducing the formation of AGEs. Phenolic compounds from sources like trans-resveratrol and hesperetin have been shown to decrease plasma levels of methylglyoxal and other 1,2-dicarbonyl compounds, while also enhancing the activity of glyoxalase 1, involved in detoxifying methylglyoxal. Moreover, Cruz et al. [45] highlighted the Mediterranean diet’s impact on 1,2-dicarbonyl compounds and AGEs, particularly through the consumption of olive oil and antioxidant-rich foods. Olive oil, with its high content of monounsaturated fatty acids and polyphenols, leads to lower postprandial levels of 1,2-dicarbonyl compounds compared to oils high in polyunsaturated fatty acids. Additionally, the diet’s emphasis on antioxidants from vegetables, fruits, and red wine helps combat oxidative stress and inflammation, potentially reducing the formation of harmful compounds like AGEs.

Certainly, numerous factors are related to glycolysis. For instance, added sugars containing fructose are found in almost 70 percent of processed foods and have been linked to the development of numerous NCDs. Chronic high intake of fructose, primarily from sources like sugar-sweetened beverages (SSBs), has been associated with various NCDs, including chronic kidney disease (CKD), chronic obstructive pulmonary disease (COPD), and asthma. Nakagawa et al. [46] reported that fructose metabolism in the kidney differs from glucose metabolism; fructose metabolism leads to the activation of pathways associated with inflammation and fibrosis. Specifically, the Warburg effect, characterized by increased glycolysis and decreased mitochondrial respiration, may be induced by fructose metabolism in CKD. Fructose metabolism generates fructose 1-phosphate (Fru1P), which promotes glycolysis over oxidative phosphorylation, contributing to intrarenal inflammation and fibrosis. This metabolic shift exacerbates renal injury, especially under pathological conditions like ischemia and high glucose levels, which stimulate endogenous fructose production. Nakagawa et al. [46] emphasized that fructose-induced inflammation involves the release of cytokines and activation of inflammatory pathways, further damaging kidney tissues. However, sodium-glucose cotransporter-2 (SGLT2) inhibitors offer a potential therapeutic strategy for CKD. By reducing glucose reabsorption in the proximal tubules, SGLT2 inhibitors indirectly decrease fructose uptake and metabolism in the kidney. Moreover, these authors highlighted that the metabolic plasticity of macrophages in response to fructose metabolism adds complexity to the role of nutrition in renal inflammation. Fructose preferentially fuels pro-inflammatory macrophages through glycolysis, exacerbating kidney damage. Oxygen availability influences macrophage metabolism, with glycolysis favored under hypoxic conditions.

Hernández-Díazcouder et al. [47] reported that fructose consumption, whether in low or high doses, stimulates uric acid (UA) production, which has been linked to impaired lung function and increased systemic inflammation. Elevated UA levels are observed in patients with acute lung injury, suggesting a potential role in lung damage. Fructose intake also affects RAGE signaling, decreasing soluble RAGE levels and potentially enhancing RAGE activity, implicated in inflammatory responses in pulmonary diseases. Moreover, these authors highlighted that fructose consumption modulates the mammalian target of rapamycin complex 1 (mTORC1) activity, a key regulator of cellular growth and metabolism. Studies indicate that fructose ingestion increases mTORC1 phosphorylation and downstream markers in various tissues, including the liver and muscles. These metabolic alterations induced by high fructose intake contribute to low-grade inflammation, which may exacerbate lung diseases.

In both asthma and COPD, nutritional factors and weight status play significant roles. Holtjer et al. [48] reported that asthma, overweight, and obesity are associated with increased risk, with higher odds ratios observed for obese individuals compared to those with a healthy body mass index (BMI). Conversely, for COPD, low BMI is identified as a risk factor, with several studies reporting elevated odds ratios for individuals with lower BMI values. While there are differences in specific risk factors between asthma and COPD, such as the direction of association with BMI, there are also notable similarities. Both conditions share risk factors, such as low birth weight, pre-term delivery, occupational dust exposure, smoking, and indoor air pollution. Additionally, Holtjer et al. [48] highlighted that dietary patterns contribute to both asthma and COPD risk. A Western-style diet high in refined grains, red and processed meat, and saturated fats is linked to increased COPD risk, while a diet high in fruits and vegetables appears to reduce the risk. Vitamin D deficiency is identified as a risk factor for COPD, suggesting the importance of adequate vitamin intake.

Similar results are reported by Nuzzi et al. [49] on pediatric asthma. They emphasized that maternal dietary patterns significantly influence immune system development and microbiome composition in newborns, impacting asthma risk. Adherence to a healthy diet rich in cooked green vegetables and Mediterranean diet principles during pregnancy correlates with reduced wheezing and asthma risk in offspring, contrasting with the increased risk associated with high meat intake or a Western diet. Supplementation of certain vitamins, notably vitamin D, vitamin A, and vitamin E, demonstrates immunomodulatory effects linked to reduced asthma risk in children. Omega-3 fatty acids from fish oil supplementation during pregnancy show the potential to decrease allergic sensitization and asthma risk. Similarly, the role of prebiotics and probiotics in shaping the microbiome and immune responses requires clearer evidence for efficacy in asthma prevention. Moreover, Nuzzi et al. [49] emphasized early-life dietary factors, including breastfeeding and timing of solid food introduction. Breastfeeding offers vital immune support, although its long-term protective effect against asthma is uncertain. Hydrolyzed infant formulas, while recommended for high-risk infants, lack consistent evidence for asthma prevention. Dietary patterns, such as the Mediterranean diet, rich in antioxidants and omega-3s, show promise in reducing asthma prevalence. Conversely, Western diets may exacerbate airway inflammation.

Indeed, several studies have focused on the Mediterranean diet’s health benefits, renowned for its antioxidant and anti-inflammatory properties. Particularly, Montano et al. [50] highlighted the Mediterranean diet as a potent shield against male infertility and cancer risks heightened by environmental pollutants. Key to this defense is represented by flavonoids, abundant in the Mediterranean diet, that counteract oxidative stress induced by pollutants, such as heavy metals, bisphenols, PAHs, dioxins, and phthalates. By scavenging free radicals, chelating toxic metals, and modulating inflammatory pathways, flavonoids mitigate cellular damage and reduce cancer susceptibility. Montano et al. [50] reported that adhering to the Mediterranean diet positively influences sperm quality, with nutrients, like omega-3 fatty acids, antioxidants, and vitamins crucial for spermatogenesis. Furthermore, flavonoid-rich foods like fruits, vegetables, nuts, and olive oil improve testicular tissue function, offering protection against sperm DNA damage. Organic cultivation practices further enhance the diet’s benefits by reducing exposure to pesticides and pollutants, thereby promoting reproductive health. Finally, Montano et al. [50] emphasized that in polluted environments where cancer and infertility rates soar, embracing the Mediterranean diet represents a vital intervention. Its antioxidant-rich components counteract sperm damage induced by environmental toxins, safeguarding male fertility amid escalating pollution levels. Flavonoids, found abundantly in the Mediterranean diet, play a pivotal role in this protection, offering a natural and effective strategy to mitigate the adverse effects of environmental pollutants on male reproductive health and overall well-being.

### 4.2. Pollutants

Münzel et al. [51] focused on urbanization, its impact on cardiovascular NCDs, and the effect of environmental factors on human health. The authors underlined that air pollution, a complex mixture of nano- to micro-sized particles (PM_10_, PM_2.5_, PM_10–2.5_, PM_0.1_, and UFP) and gaseous pollutants (NO_2_, SO_2_, CO, and O_3_) in the urban environment, is the leading environmental risk factor for global health and the fourth largest risk factor for global mortality. Moreover, they considered outdoor air pollution exclusively, although indoor air pollution also represents a risk factor for respiratory and cardiovascular NCDs [51].

In their umbrella review, Holtjer et al. [48] reviewed 75 studies on COPD and adult-onset asthma (AOA), identifying 43 and 45 risk factors, respectively. BMI emerged as one of the main differences, with a high BMI posing a risk factor for AOA, whereas a low BMI increased the risk for COPD. Additionally, gender did not show any disparities in AOA risk, whereas males exhibited a higher susceptibility to COPD. Furthermore, for both diseases, the authors evidenced that low birth weight and pre-term delivery were implied as risk factors, although pre-term delivery was not statistically significant for either. Occupation dust exposure was considered in different reviews included in the umbrella review for both conditions, highlighting its significance in prevention efforts. Active and passive smoking were unsurprisingly identified as crucial risk factors for both conditions. Lastly, ambient air pollution and both pesticide exposure and exposure to workplace irritants were found to be risk factors for both COPD and AOA [48].

In the evaluation of environmental exposure, Kumar et al. [42] highlight the burden of human exposure to EDCs to understand the magnitude of the exposome on human health. EDCs are linked to various health issues, including fertility problems, diabetes, obesity, metabolic disorders, thyroid homeostasis, and increased risk of hormone-sensitive cancers. Common exposure sources include plastics and pesticides, and even low doses during critical developmental periods can impact the endocrine system. The authors conclude that an extensive exposome approach will facilitate the assessment of deleterious health effects of EDCs in humans through targeted biomarkers. Climate change urges countries to create laws and guidelines for safe drinking water [42].

The impact of social, psychological, and socioeconomic factors on the major NCDs. was evaluated in the narrative review of Skýbová et al. [14]. In particular, the review aimed to gather knowledge on the distribution, determinants, clusters, and psychological and socioeconomic consequences of NCDs, to demonstrate the opportunity for future health system transformation in terms of NCD prevention for public health. In this perspective, it is crucial to consider data on the exposome—which is totally lacking in epidemiological studies—as well as its impact on health in the context of NCDs. Indeed, the authors’ findings confirmed that the currently available data do not capture the heterogeneity of the existing interplay between environment, health, and differences at the population level [14].

Shi et al. [52] reviewed the recent epidemiological evidence about the correlation between exposure to various outdoor and indoor air pollutants, mainly PM, NO_x_, O_3_, and polycyclic aromatic hydrocarbons (PAHs) and overweight and obesity outcomes. Although evidence suggests a relationship between air pollutants, PM, and the risk of overweight/obesity during different life stages, the exact mechanisms linking air pollution exposure to obesity remain partially unclear. Several recent studies, evaluated in the review, have shed light on the underlying mechanisms connecting air pollution to obesity. When air pollutants enter the lungs, the respiratory system activates inflammatory cells and releases large quantities of biological intermediates that contribute to lung inflammation and harm multiple organ systems via blood circulation. PM_2.5_ and ultrafine particles (UFPs) also enter the bloodstream directly through the air–blood barrier and are distributed to secondary target organs, where they exert harmful effects. Additionally, PAHs, which are widespread organic pollutants, have been implicated in endocrine disruption and carcinogenesis. Furthermore, exposure to PAHs during pregnancy has been associated with childhood obesity [52].

In discussing COPD, Elonheimo et al. [53] suggest that beyond traditional risk factors, like smoking, some environmental substances can increase the risk of disease. Pesticides can lead to both acute and chronic respiratory issues, with evidence linking them to COPD. Moreover, occupational exposure to pesticides has been linked to both COPD and chronic bronchitis. In particular, cadmium (Cd) mainly affects the kidneys but also impacts bones and the respiratory (via inhalation), endocrine, and reproductive systems; Chromium (Cr) from air pollution showed a strong association with respiratory and COPD mortality, and Cr and Cr (VI) were adversely associated with lung function. Both short and long-term exposure to arsenic (As) pose serious health risks, potentially decreasing lung function, although its direct link to COPD risk remains uncertain. Lead (Pb) is a human carcinogen and in both adults and children; Pb exposure was found to be associated with decreased lung function and increased respiratory diseases including COPD. Diisocyanates can cause harmful health effects; exposure to them often leads to occupational asthma globally, but less attention has been given to COPD or related symptoms. Evidence suggests a link between diisocyanate exposure and pulmonary changes; particularly, toluene diisocyanate exposure affects airway caliber, epithelial permeability, and lung function. Moreover, exposure to PAHs is linked to reduced lung function and respiratory problems like COPD and bronchitis [53].

Montano et al. [50] provided a comprehensive review of the role of the Mediterranean diet and, in particular, the consumption of flavonoids for their beneficial effects on human health, including the mitigation of risks derived from environmental pollutants. It highlights their role in counteracting male infertility and cancer risk induced by environmental pollutants, such as heavy metals, bisphenols, PAH, dioxins, and phthalates. The authors claim that, despite promising results from current research, further studies are necessary to definitively assert the role of flavonoids in the protection of fertility and associated cancers caused by exposure to environmental pollutants. This would allow the development of novel evidence-based dietary interventions for protecting the health of individuals living or working in highly polluted environments [50]. Instead, the role of vitamins and dietary patterns in the epigenetic aspect of NCDs, the mechanisms of action of vitamins in epigenetic modification, and the effects of vitamin deficiency on changes in epigenetic patterns were described in Khajebishak’ et al. [43] review paper. Epigenetic phenomena are largely determined by DNA methylation, chromatin remodeling, histone modification, and long non-coding RNAs. For this reason, it is crucial to adhere to a healthy dietary pattern with sufficient vitamins and avoid consuming a Western diet. Consuming the recommended dietary level of vitamins and maintaining a healthy diet can contribute to epigenetic stability [43].

Filho et al. [54] reviewed the literature on air pollution and indoor settings. Among NCDs, household air pollution (HAP) from polluting fuels causes 25% of stroke deaths, 15% of heart disease deaths, 17% of lung cancer deaths, and over 33% of COPD deaths. Exposure to pollutants triggers oxidative stress and inflammatory signaling which increase as the surface area of ultrafine particles rises. Air pollutants can induce oxidative stress, inflammatory responses, and respiratory diseases. Genetic variations in glutathione S-transferase genes influence susceptibility to pollution-induced respiratory and allergic diseases.

Indeed, exposure to NO_2_, PM_2.5_, and PM_10_ during pregnancy and the first year of life was associated with a higher prevalence of abnormal lung function and the development of asthma in children. Moreover, air pollution increases cardiovascular disease and shortens life expectancy. It may also cause eye irritation, cataracts, dry eye disease, and blepharitis. It is carcinogenic and can affect skin quality and aging [54].

HAP exposure is believed to be linked to respiratory diseases, and current evidence indicates that exposure to indoor air pollution due to biomass smoke is possibly strongly associated with COPD [54].

The findings of all the selected articles on exogenous factors are listed in Table 2.

## 5. Role of Environmental Factors on NCDs

The exposome represents all the types of environmental exposure and pathogens to which every person has been exposed since conception. It allows us to understand how the totality of an individual’s exposure over the course of life can affect human health. All forms of pollution could have a potential impact on human health. In this regard, the exposome provides a complete description of the history of exposure throughout life.

Air pollution has been identified as the main health risk factor, when contrasted with water and soil pollution due to heavy metals, pesticides, other chemicals, and occupational exposure, but non-chemical aspects were neglected in [55].

Air pollution is divided into two categories: external environmental pollution includes residential and commercial energy use, agricultural emissions, fossil fuels for energy production and biomass fuels, industrial sources, and traffic-related air pollution; internal environmental pollution derives from the spread of external air pollution and indoor activities (stoves, cooking, detergents, tobacco smoke, dust, microbes, allergens, building materials, furniture, paints, floors, and paper wallpaper) [52].

### 5.1. External Environmental Pollution

#### 5.1.1. Obesity and Cardiometabolic Diseases

Recently, air pollution has been associated with obesity, and obesity is associated with respiratory diseases and is a risk factor for cardiovascular diseases such as myocardial infarction and hypertension and for the development of metabolic complications, particularly type 2 diabetes and hepatic steatosis. The risk of breast, ovarian, prostate, kidney, liver, and colon cancers are also significantly increased in obese individuals. Additionally, obesity is associated with cognitive impairment and dementia, Alzheimer’s disease, low self-esteem, and depression. In addition to already-known factors, such as genetic and behavioral ones, the current obesity epidemic is also driven by environmental factors, in particular air pollution. Many biomarkers, including inflammatory cells, cytokines, and C-reactive protein (CRP), have been used to reflect the inflammatory state, first pulmonary then systemic, induced by exposure to air pollution. Subsequently, the activation of systemic inflammation leads to the recruitment of pro-inflammatory cells and cytokines into adipose tissue and other tissues, provoking a local inflammatory response. Inflammation of adipose tissue can lead to metabolic disorders and obesity. Oxidative stress caused by exposure to air pollution is also related to obesity. Furthermore, air pollution causes endocrine interference, which induces metabolic disorders through the production of adipokines (leptin, adiponectin, and pro-inflammatory cytokines) produced by adipose tissue which can lead to obesity [52]. Many chronic diseases with an inflammatory component show significantly increased levels of electrophiles. The main function of the mercapturate pathway is to detoxify electrophilic species. Electrophiles that could arise from the metabolism of endogenous or exogenous substances, or their biotransformation products, are present in the air, food, or water. Most of the knowledge on the mercapturate pathway has been provided by environmental biomonitoring studies exposed to toxic substances. Endogenous metabolites related to the mercapturate pathway formed in humans (the mercapturomic profile) are key pathophysiological factors in the onset and development of chronic non-communicable inflammatory diseases, including cardiometabolic diseases (in particular diabetes, atherogenesis, or obesity related to OSAS (obstructive sleep apnea syndrome), intracerebral hemorrhage, and coronary heart disease) [56].

#### 5.1.2. Cardiovascular Diseases

Air pollution, in the form of solid particles or reactive gases, such as ozone and nitrogen dioxide, contributes to the development and progression of cardiovascular and cerebrovascular diseases, as well as cardiovascular mortality. In particular, PM_2.5_ is a precursor of cardiovascular diseases with a constant increase in deaths due to air pollution. Even environmental noise, traffic in particular, significantly affects the development/progression of cardiovascular diseases and can contribute to metabolic diseases. Both chronic and acute exposure cause endothelial dysfunction, oxidative stress, inflammation, and activation of pro-thrombotic pathways, which are crucial for the induction of adverse effects. Morbidity and mortality due to pollution occur more frequently in young (<5 years) and elderly (>60 years), especially in low- and middle-income countries. The precise mechanisms leading to noise-induced vascular damage at the molecular level and subsequently to cardiovascular disease are only poorly protected from a mechanistic point of view and the perspective of particular biomarkers [55].

#### 5.1.3. Cellular Aging

The search for biomarkers of age, such as telomere length, epigenetic modifications, markers of DNA damage, inflammation, oxidative stress, and indicators of mitochondrial function have emerged as valuable tools for assessing the effects of environmentally toxic substances on aging. Evaluating the effects of environmental toxicants on aging through biomarkers allows us to identify individuals at greater risk of accelerated aging or age-related diseases [31]. The inflammatory environment, promoted by prolonged exposure to these environmental pollutants, is associated with aging as it increases oxidative stress, and DNA damage, induces cellular senescence, and causes dysfunction of stem and mitochondrial cells. By capturing the totality of environmental exposure over an individual’s lifetime, the exposome approach provides a holistic perspective on the cumulative effects of this exposure on aging processes [31].

#### 5.1.4. Dementia and Neurodegenerative Diseases

Air pollution, particularly PM_2.5_, is implicated in the pathogenesis of both Parkinson’s and Alzheimer’s diseases and related dementias [31]. These chronic non-communicable inflammatory diseases, as well as autism, show increased levels of electrophiles; therefore, endogenous metabolites related to the mercapturomic profile could represent a useful tool to characterize the relationship with the disease [56].

#### 5.1.5. Other Pathologies

Air pollution is associated with circulatory, respiratory, neurological, and metabolic diseases, birth defects, and premature births. Environmental factors impact the early stages of life, from the embryo, and can represent the origin of adult disease or exert a beneficial effect on long-term health [52]. Mental stress, exposure to light, climate change, and traffic noise can also contribute to the onset of chronic non-communicable diseases and have effects on the exposome [55]. Air pollution, especially PM_2.5_, is considered a significant risk factor for respiratory disorders, such as COPD. Additionally, individuals living in areas with higher concentrations of PM_2.5_ particles have been found to have a greater risk of osteoporotic fractures. There is also a potential link between air pollution and arthritis. Air pollution is also associated with lung cancer. Exposure to ultraviolet (UV) radiation is a major risk factor for most melanomas [31]. Increased levels of electrophiles are found in melanoma, non-Hodgkin’s lymphoma, breast, ovarian, and thyroid cancers. The studies examined quantify only one type or family of metabolites related to the mercapturate pathway, capable of detoxifying electrophiles associated with the disease (dopamine, estrogens, cysteinyl-leukotrienes, and cysteinyl-S-conjugates), and therefore, qualitative analyses in the future and complete quantitative analyses of the mercapturomic profile could represent a useful tool to characterize the relationship with the disease [56].

### 5.2. Internal Environmental Pollution

While outdoor air pollution is widely discussed, indoor air pollution, including that from domestic activities and workplaces, is equally crucial, underscoring its impact on non-communicable respiratory and cardiovascular diseases [51]. People spend most of their time indoors. Indoor air pollution includes various biological contaminants such as allergens, mainly house dust mites, insects, pollen, or other substances deriving from animals, molds, and bacterial endotoxins. Other components are chemical air pollutants, such as gases, particulate matter, formaldehyde, and volatile organic compounds. Chemical environmental pollutants are associated with cardiovascular disorders (myocardial infarction, stroke, heart failure, and increased risk of mortality), ocular problems (conjunctivitis, cataracts, dry eye, and blepharitis), respiratory problems (acute respiratory infections, tuberculosis, asthma, COPD, and pneumoconiosis), airways (allergic rhinitis and asthma), cancer (lung, colorectal, stomach, kidney, and bladder), and premature skin aging (wrinkles) [54].

### 5.3. Water

The total of external pressures from the water-related exposome is a significant determinant of human health. Domestic, industrial, and agricultural emissions in water lead to a polluted state, with thousands of substances and their residues including nutrients, heavy metals, pesticides (insecticides, herbicides, and fungicides), and pharmaceutical products. The impact on health is direct, through exposure to polluted water and the consumption of drinking water, bathing, swimming, or inhaling, but also indirect, through the ecosystem. Due to chemical water pollution, acute poisoning occasionally occurs, but more often, the effects take years to manifest, causing birth defects and incurring other long-term health effects. The accumulation of confounding factors over time hinders specific attribution. Furthermore, water is microbiologically contaminated and is a habitat for vector-borne diseases and has potential cumulative effects from plastic debris. Plastic waste in water poses health risks through particle and chemical toxicity. Exposure to microplastics can contribute to oxidative stress, inflammation, and endothelial dysfunction, which can increase the risk of cardiovascular disease [57]. Pollutants and contaminants—such as arsenic, lead, and cadmium present in drinking water—and mercury, can increase the risk of cardiovascular diseases (hypertension, atherosclerosis and cardiovascular mortality, hypertension, coronary heart disease, and myocardial infarction). Exposure to lead in contaminated water leads to a decrease in bone mineral density and an increased risk of fractures [31].

The concept of the exposome refers to the totality of exposure to a variety of external and internal sources, including chemical agents, biological agents, or radiation, over a lifetime. To date, environmental epidemiology has focused primarily on difficult outcomes, such as mortality, disease exacerbation, and hospital admissions. However, our understanding of the genesis of complex non-communicable diseases linked to environmental exposure will benefit from a better understanding of biomarkers of exposure and the exposome [51].

The characteristics of all the included articles on the role of environmental factors on NCDs are listed in Table 3.

## 6. Conclusions

Prevention and management of NCDs requires a thorough understanding of the interactions between exogenous and endogenous factors. Environmental factors such as pollutants, diet, air, and water quality, along with genetic and microbiota influences, play a crucial role in the risk and development of NCDs. A One Health approach, which recognizes the interconnectedness of human, animal, and environmental health, is essential for addressing these complex challenges. This integrated approach considers how the health of humans is closely linked to the health of animals and the environment in which they live. Preventive and therapeutic strategies that consider these multiple interactions can reduce the risk of NCDs and promote sustainable global health.

The exposome’s role in non-communicable chronic diseases is illustrated in Figure 2.

## Figures and Tables

**Figure 1 diseases-12-00176-f001:**
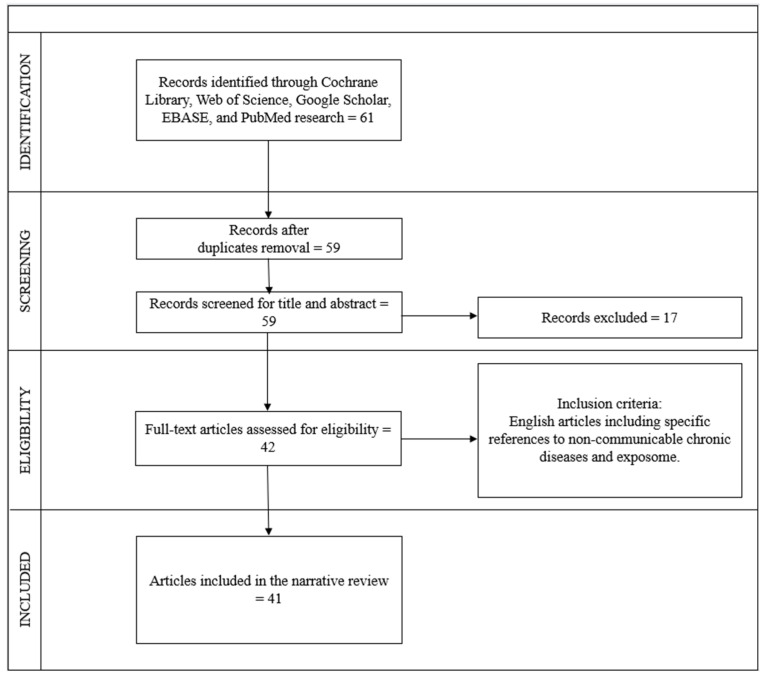
Flow diagram of study selection, according to PRISMA and REAPPRAISED checklist methods.

**Figure 2 diseases-12-00176-f002:**
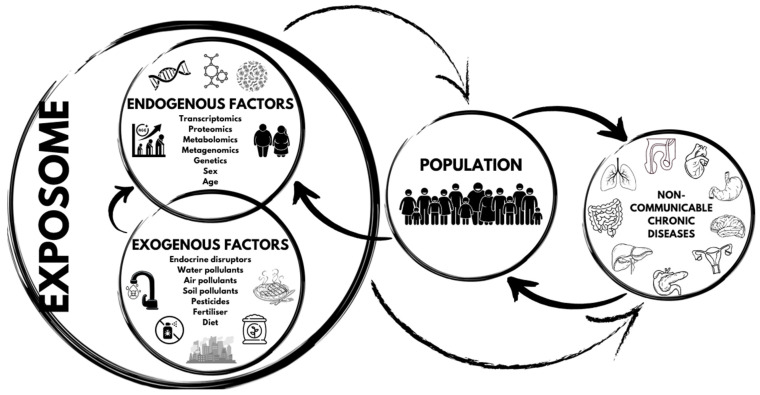
The exposome’s role in non-communicable chronic diseases.

**Table 1 diseases-12-00176-t001:** Summary of characteristics of included studies on endogenous factors in the most recent reviews, systematic reviews, and meta-analyses.

Factors	Finding	References
Genetics and epigenetics	The exposome, encompassing environmental chemicals and food antigens, influences health and autoimmune disease through genetic interactions. Understanding diet’s impact and synthetic chemicals from occupational exposure and smoking is critical for prevention. Improving habits to reduce exposure to environmental pollutants can mitigate tissue damage and autoimmune responses by modulating immune cells.	Vojdani, A.; et al. [17]
Genetics and epigenetics	Environmental toxicants induce oxidative stress, DNA damage, alter DNA methylation, and affect gene expression, contributing to varied autoimmune disease presentations. Genetic predisposition and environmental factors like lifestyle changes and antibiotic use in food alter gut microbiota diversity, impacting immune function and triggering pro-inflammatory pathways in autoimmune disorders.	Martinelli, S.; et al. [18]
Genetics and epigenetics	Nutri-epigenetics explores how dietary components influence gene expression via epigenetic mechanisms like histone modification and DNA methylation.	Gabbianelli, R.; et al. [19]
Genetics and epigenetics	Promoter methylation affects gene transcription and protein production by reducing interactions with transcription factors and RNA-polymerase. Maternal lifestyle, diet during pregnancy, and exposure to environmental pollutants influence epigenetic mechanisms and microbiota development in infants, potentially impacting long-term health outcomes like inflammatory responses and DNA damage linked to carcinogenesis.	de Oliveira Alves, N.; et al. [20]
Microbiota	The microbiota influences the immune system, maintaining a balance crucial for health. Dysbiosis, an imbalance in gut microbial composition, contributes to diseases through various pathways influenced by factors like diet, antibiotics, and inflammation. This interaction is pivotal in gastrointestinal cancer development and underscores the bidirectional modulation between host immunity and microbiota at the gut interface.	Cianci, R.; et al. [21]
Microbiota	Environmental pollutants and climate change disrupt microbiota, favoring pathogens over beneficial bacteria, causing gut inflammation and increased disease risk. Air, water, and land pollution induce widespread inflammation and oxidative stress, impacting immunity and potentially compromising vaccination effectiveness.	Franza, L.; et al. [22]
Microbiota	Human cells in the body (10^13^) are outnumbered by bacterial cells (10^14^) in the microbiome, which can be influenced by dietary components and xenobiotics, affecting various physiological processes. The gut environment, shaped by factors like oxygen levels and pH, influences microbial diversity, impacting immune and neuroendocrine responses, with diet and external factors modulating this complex system and its role in health and disease.	Moon, Y.; et al. [23]
Microbiota	The rise in allergic diseases is linked to genetic and environmental factors, with modern changes leading to epithelial barrier dysfunction and microbial dysbiosis. Maintaining tissue homeostasis relies on epithelial cells, but environmental pollutants and aging can compromise these barriers, increasing susceptibility to inflammation and chronic diseases.	Losol, P.; et al. [24]
Microbiota	Environmental factors, particularly changes in diet and exposure to pollutants, are increasingly recognized as dominant causes of allergic diseases through the exposome’s impact on the epithelial barrier, leading to inflammation and microbial imbalance. The shift towards processed foods, high in omega-6 fatty acids and low in antioxidants, along with environmental pollutants, has been linked to rising rates of allergies and chronic inflammatory diseases.	Celebi Sozener, Z.; et al. [25]
Microbiota	The epithelial barrier hypothesis suggests that environmental toxins and genetic factors compromise epithelial barriers, leading to chronic diseases such as allergies, autoimmune, and metabolic disorders. Industrialization, urbanization, and modern lifestyles disrupt these barriers, causing inflammation and microbial dysbiosis.	Pat, Y.; et al. [26]
Microbiota	The microbiome, considered an organ, is crucial in host functions, influencing cancer development and treatment by affecting inflammation and microbial balance. Chronic inflammation, often due to infections, contributes to cancer progression by altering the tumor microenvironment and immune responses, with both commensal and pathogenic microorganisms playing significant roles.	Jabłońska-Trypuć, A.; et al. [27]
Microbiota	Epigenetics and nutrigenomics reveal how environmental factors and genetic variations interact to influence gene expression and metabolism, impacting disease susceptibility and dietary responses. Understanding these interactions can guide personalized nutrition strategies, highlighting the importance of genetic testing and tailoring dietary interventions for optimal health outcomes.	Bordoni, L.; et al. [28]
Microbiota	CRC results from genetic–environmental interactions involving lifestyle factors and gut microbiota-derived metabolites, which cause inflammation, DNA damage, and metabolic issues.	Zhang, W.; et al. [29]
Aging	Aging is a complex biological process leading to a decline in physical function due to accumulated damage from various stressors, with genetic, epigenetic, and environmental factors playing key roles.	Guo, J.; et al. [30]
Aging	The aging global population is seeing an increase in the prevalence of NCDs, including Alzheimer’s, Parkinson’s, and COPD, exacerbated by environmental toxicants contributing to 24% of global deaths. Additionally, environmental pollutants and occupational exposure are significant risk factors for various cancers.	Pandics, T.; et al. [31]
Aging	Inflammageing, driven by senescent cells and inflammatory markers like IL-6 and CRP, complicates age-related disease understanding. Interventions targeting mTOR, such as rapamycin and metformin, improve health span by suppressing inflammageing through autophagy activation and cytokine modulation.	Teissier, T.; et al. [32]
Aging	Heavy metals like lead and aluminum accelerate human aging, impacting neurological diseases such as Alzheimer’s and Parkinson’s, while limited studies suggest heavy metals’ associations with epigenetic changes and miRNA expression. Various types of chemical exposure from addictions (e.g., alcohol, cocaine, and nicotine) and occupational hazards (e.g., solvents) also influence aging, with smoking linked to skin aging and cognitive decline, underscoring diverse impacts on health span.	Misra, B.B.; et al. [33]
Aging	Exposure to air pollution, including both outdoor (e.g., ozone and particulate matter) and indoor sources (e.g., solid fuel combustion), accelerates skin aging by promoting pigment spots and wrinkles. Mechanistic and epidemiological studies highlight synergistic effects between UV radiation and pollutants like PM, emphasizing oxidative stress and genetic damage as underlying mechanisms.	Schikowski, T.; et al. [34]
Aging	The Mediterranean diet emphasizes a high consumption of cereals, legumes, fruits, and vegetables. These foods have the potential to reduce chronic disease risks and improve longevity through mechanisms such as telomere maintenance and modulation of oxidative stress. Its impact on telomere length varies depending on the population and genetic factors.	Davinelli, S.; et al. [35]
Obesity	The exposome concept examines a broad spectrum of human environmental exposure, encompassing urban settings, chemicals, lifestyles, and social factors, providing a comprehensive view of health impacts. With causal mediation and quantile g calculation, the relationships between environmental factors and the influence of socioeconomic status on birth weight were studied through the exposome, highlighting complex sets of mediators and interventional effects.	Maitre, L.; et al. [36]
Obesity	The concept of the exposome encompasses all environmental factors that impact human health, highlighting the need for a holistic approach that goes beyond individual exposure such as air pollution or pesticides. Different types of environmental exposure during early life result in molecular changes (e.g., methylome, transcriptome, and metabolites), identifying potential biomarkers and mechanisms underlying disease susceptibility.	Maitre, L.; et al. [37]
Obesity	Childhood obesity rates are rising globally. Higher BMI and childhood adiposity correlate with increased risks of type 2 diabetes, cardiovascular diseases, cancers, poor academic performance, and mental health issues. Environmental factors like chemical contaminants (e.g., persistent organic pollutants and metals), urban settings, and lifestyle influence obesity through complex mechanisms.	Vrijheid, M.; et al. [38]
Obesity	The global obesity epidemic, exacerbated by sedentarism and longer life expectancy, presents a significant public health challenge. Comprehensive analyses reveal environmental drivers like physical inactivity and endocrine disruptors, advocating for an exposome approach to understand obesity’s complex global spread and impacts.	Catalán, V.; et al. [39]
Obesity	Tryptophan undergoes metabolism via kynurenine and serotonin pathways or by gut microbes, producing AHR ligands such as kynurenine and indole derivatives that impact immune responses and inflammation. In IBD, genetic factors, lifestyle, diet, and the microbiota are contributors; reduced serum tryptophan levels correlate with disease activity, while microbiota-derived metabolites modulate IBD severity via AHR activation.	Dong, F.; et al. [40]
Obesity	The gut microbiota produces different metabolites influencing host physiology and disease, including AHR ligands from tryptophan metabolism, crucial for immune regulation. Impaired AHR ligand production correlates with reduced GLP-1 secretion, linking microbiota, AHR signaling, and metabolic syndrome pathogenesis.	Natividad, J.; et al. [41]

Abbreviations: AHR, aryl hydrocarbon; BMI, body mass index; COPD, chronic obstructive pulmonary disease; CRC, colorectal cancer; CRP, C-reactive protein; DNA, deoxyribonucleic acid; GLP-1, glucagon-like peptide-1; IBD, inflammatory bowel disease; IL, interleukin; mTOR, mechanistic target of rapamycin; NCDs, non-communicable diseases; PM, particulate matter; RNA, ribonucleic acid; UV, ultraviolet.

**Table 2 diseases-12-00176-t002:** Summary of findings of the included studies on exogenous factors in the most recent reviews, systematic reviews, and meta-analyses.

Factors	Finding	Author
Diets; Pollutants; EDCs	Environmental endocrine-disrupting chemicals known as EDCs, including plastics and pesticides, pose significant health risks by disrupting hormone systems, potentially causing developmental, reproductive, metabolic, and neurobehavioral disorders. These chemicals accumulate in the environment and human body, impacting health across generations through various exposure routes.	Kumar, M.; et al. [42]
Diets; Vitamins	NCDs like diabetes, cardiovascular diseases, and cancer are major global health challenges exacerbated by factors such as unhealthy diets, physical inactivity, air pollution, and vitamin deficiencies. Epigenetic modifications influenced by dietary patterns and micronutrient deficiencies play crucial roles in the pathogenesis of NCDs.	Khajebishak, Y.; et al. [43]
Diets; AGEs	AGEs, formed from sugar–amino group reactions, impact health via protein dysfunction and RAGE activation. In vitro studies on dietary AGEs show reduced digestibility and varied effects on cell proteins, contrasting with inconclusive in vivo results on intestinal health pathways and mechanisms.	Jansen, F. A. C.; et al. [44]
Diets; AGEs; 1,2-Dicarbonyl Compounds	1,2-Dicarbonyl compounds like methylglyoxal and glyoxal are electrophilic molecules from metabolism and diet, linked to AGEs in diseases. Mediterranean diets, rich in fiber, can modulate their levels, potentially reducing their harmful effects through complex postprandial dynamics.	Cruz, N.; et al. [45]
Diets; Fructose	Fructose-induced inflammation resembling the Warburg effect exacerbates CKD, explaining SGLT2 inhibitors’ protective role via fructose metabolism modulation across diabetic and non-diabetic CKD. Recent evidence suggests diabetes suppresses mitochondrial function, promoting glycolysis in diabetic nephropathy. SGLT2 inhibitors may mitigate renal inflammation and fibrosis by blocking this metabolic shift in CKD.	Nakagawa, T.; et al. [46]
Diets; Fructose; AGEs	Chronic high fructose intake from sugar-sweetened beverages in LATAM links to obesity, diabetes, cardiovascular diseases, and possibly lung diseases like COPD and asthma, mediated through uric acid, inflammatory pathways, RAS, AGEs, and RAGE signaling, suggesting fructose worsens conditions like acute lung injury and pulmonary fibrosis and potentially impacts COVID-19 severity.	Hernández-Díazcouder, A.; et al. [47]
Diets; Pollutants; Air pollution; Pesticides	COPD and childhood asthma are linked to shared risk factors such as air pollution and exposure in the early years of life. A higher BMI increases the risk of asthma in adulthood but reduces the risk of COPD, while smoking and exposure to dust in the workplace are common risks.	Holtjer, J. C. S.; et al. [48]
Diets; Air pollution	Childhood asthma, a prevalent chronic condition, results from complex interactions among genetic, epigenetic, and environmental factors, including prenatal and early-life exposure like tobacco smoke, air pollutants, and dietary patterns. There is a potential protective role of maternal consumption of specific foods (e.g., cooked green vegetables), adherence to Mediterranean diets, and omega-3 fatty acids intake during pregnancy.	Nuzzi, G.; et al. [49]
Diets; Pollutants; Vitamins; EDCs;	Environmental pollutants, especially endocrine disruptors, impact male fertility and increase testicular cancer risks, but the Mediterranean diet, rich in antioxidants like flavonoids (e.g., rutin and quercetin), could mitigate these effects by combating oxidative stress and inflammation, thereby improving reproductive health outcomes.	Montano, L.; et al. [50]
Pollutants; Air pollution	Urban environments significantly contribute to air pollution, exacerbating NCDs like cardiovascular disease and diabetes. Mitigation strategies for alleviating the associated health risks globally include urban planning reforms, noise barriers, and reducing exposure to both transportation noise and light pollution.	Münzel, T.; et al. [51]
Pollutants; EDCs; Air pollution; PM	Obesity, a global crisis exacerbated by air pollution, particularly PM_2.5_, impacts health differently across genders due to endocrine disruptions and sex hormone interactions. Understanding these disparities and their underlying mechanisms is crucial for targeted prevention strategies aimed at mitigating obesity risks associated with environmental factors.	Shi, X.; et al. [52]
Pollutants; Air pollution; Pesticides; Metals	COPD is linked to irreversible airflow obstruction and comorbidities like emphysema and chronic bronchitis. Environmental exposure to pesticides, cadmium, lead, arsenic, diisocyanates, and polycyclic aromatic hydrocarbons is associated with COPD, impacting lung function and exacerbating disease severity.	Elonheimo, H. M.; et al. [53]
Pollulants; Air pollution; Indoor air pollulants	Indoor air pollution, driven by nonbiological contaminants in developing countries, leads to substantial health risks. This pollution causes millions of premature deaths annually, mainly from NCDs, notably impacting vulnerable groups like women and children due to high exposure levels and their susceptibility to respiratory issues.	Rosário Filho, N. A.; et al. [54]

Abbreviations: AGEs, advanced glycation end product; CKD, chronic kidney disease; COPD, chronic obstructive pulmonary disease; EDCs, endocrine-disrupting chemicals; LATAM, Caribbean and Latin America; NCDs, non-communicable diseases; PM, particulate matter; RAGE, receptor for advanced glycation end products; RAS, renin-angiotensin system; SGLT2, sodium/glucose cotransporter 2.

**Table 3 diseases-12-00176-t003:** Summary of characteristics of the included studies on the role of environmental factors on NCDs in the most recent reviews, systematic reviews, and meta-analyses.

Author	Title	Type of Paper	Date	Finding
Daiber, A.; et al. [55]	The “exposome” concept—how environmental risk factors influence cardiovascular health.	Review	2019	NCDs, driven by environmental factors like air pollution and noise, highlight the exposome’s role in lifelong health. PM_2.5_ contributes to cardiovascular mortality, while transportation noise worsens cardiovascular health and impacts children’s cognition and sleep.
Gonçalves-Dias, C.; et al. [56]	The mercapturomic profile of health and non-communicable diseases.	Review	2019	The mercapturate pathway in renal proximal tubular cells and hepatocytes detoxifies electrophilic species from endogenous and exogenous sources by forming Cys-S-conjugates extracellularly. These Cys-S-conjugates, known for their longer half-life and toxicity potential, play roles in various diseases like neurologic disorders and cardiometabolic diseases.
Boelee, E.; et al. [57]	Water and health: From environmental pressures to integrated responses.	Review	2019	Water-related environmental pressures affect human health through communicable and non-communicable diseases linked to water contamination and ecosystem changes. Urgent support for integrated water management is crucial for mitigating these health risks and fostering sustainable development, especially amidst agricultural intensification and infrastructure development that heighten disease vectors and health disparities in disadvantaged communities.

Abbreviations: Cys, cysteine; NCDs, non-communicable diseases; PM, particulate matter.

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
