# Peer review of "Exploring the Exposome Spectrum: Unveiling Endogenous and Exogenous Factors in Non-Communicable Chronic Diseases"

_diseases, 2024, doi:10.3390/diseases12080176_

Round 1
Reviewer 1 Report
Comments and Suggestions for Authors
The authors provide a review of reviews associated with the exposome. The study is comprehensive and generally well-written. However, there are several improvement areas and it is recommended the authors perform a thorough proofreading effort. Specific comments are listed below.
Line 41: "encounter" should be "encounters".
Line 82: "Review" should not be capitalized.
Line 83: "responsibility" should be "responsibilities".
Line 90: "such as" or exactly those keywords? This needs to be more specific.
Lines 90-4: There is are unnecessary spaces after the " symbols before "non".
Line 105: What does "excluded for abstracts" mean?
Figure 1: It is unclear how the number of articles went from 42 in the Eligibility part to 41.
Line 120: What does "are largely spread" mean?
Line 163: "site" should be "sites".
Line 167: Is "capability to answered" meant to be "capability to answer"?
Lines 238-9: Extreme temperatures (particularly heat) are also important environmental risk factors.
Line 251: "2.5" in "PM2.5" should be subscripted.
Table 1: The last review (Natividad et al.) does not have citation.
Line 423: The cited author should be "Jansen" not "Jansens".
Lines 549-50 (and elsewhere): Please subscript the numbers on pollutants (e.g., "10" in "PM10").
Line 575: "Countries" should not be capitalized.
Line 602: The start of the sentence does not read gramatically correct.
Line 641: It is unclear what is meant by "smaller particles entitled to produce more oxidative stress."
Table 2: None of the articles listed have an included citation.
Line 731: "exosome" seems to be "exposome".
Figure 2 could be improved as it looks to be of low resolution.
Comments on the Quality of English LanguageThe English Language of the article is of good quality. It is recommended the authors pay attention to specific comments as well as perform a thorough proofreading before final publication.
Author Response
Dear Editor of Diseases
First, my coauthors and I would like to thank You sincerely for this opportunity of cooperation. We profoundly thank the reviewers for the comments and useful suggestions aimed at improving the paper. We thank You for your constructive critique and we hope the review process has led to an improved manuscript. If additional changes are warranted, we will make them.
We hope that this revised version of our manuscript may now be found suitable for publication.
This is a point-by-point list of changes made in the paper:
Reviewer 1
“The authors provide a review of reviews associated with the exposome. The study is comprehensive and generally well-written. However, there are several improvement areas and it is recommended the authors perform a thorough proofreading effort. Specific comments are listed below.
Line 41: "encounter" should be "encounters".
Line 82: "Review" should not be capitalized.
Line 83: "responsibility" should be "responsibilities".”
- The authors thank the Reviewer. Issues highlighted in lines 41-82-83 have been fixed in the text.
“Line 90: "such as" or exactly those keywords? This needs to be more specific.”
- The authors thank the Reviewer. The sentence in lines 90-6 has been rephrased by specifying search terms.
“Lines 90-4: There is are unnecessary spaces after the " symbols before "non".”
- The authors thank the Reviewer. Issues highlighted in lines 90-4 have been fixed in the text.
“Line 105: What does "excluded for abstracts" mean?
Figure 1: It is unclear how the number of articles went from 42 in the Eligibility part to 41.”
- The authors thank the Reviewer. The sentence in line 105 and Figure 1 have been modified to specify more about the search process.
Line 120: What does "are largely spread" mean?
- The authors thank the Reviewer. The sentence in line 120 has been modified in “are largely distributed in the environment”
“Line 163: "site" should be "sites".
Line 167: Is "capability to answered" meant to be "capability to answer"?”
- The authors thank the Reviewer. Issues highlighted in lines 163-167 have been fixed in the text.
“Lines 238-9: Extreme temperatures (particularly heat) are also important environmental risk factors.”
- The authors thank and agree with the Reviewer. Therefore, the sentence in lines 238-9 has been changed to “Tobacco smoking, extreme temperatures, particularly heat, and air, water, and soil pollution are among the most important environmental risk factors for cardiovascular death.”.
“Line 251: "2.5" in "PM2.5" should be subscripted.
Table 1: The last review (Natividad et al.) does not have citation.
Line 423: The cited author should be "Jansen" not "Jansens".”
- The authors thank the Reviewer. Issues highlighted in lines 251-423 and Table 1 have been fixed in the text.
“Lines 549-50 (and elsewhere): Please subscript the numbers on pollutants (e.g., "10" in "PM10").”
- The authors thank the Reviewer. Issues highlighted in lines 549-50 have been fixed in the text, as well for lines 588-597-645-710-734-745-747 and for Tables 2-3.
“Line 575: "Countries" should not be capitalized.”
- The authors thank the Reviewer. Issues highlighted in line 575 have been fixed in the text.
“Line 602: The start of the sentence does not read gramatically correct.”
- The authors thank the Reviewer. The sentence in line 602 has been modified in “Elonheimo et al. [53] suggest about COPD that […]”
“Line 641: It is unclear what is meant by "smaller particles entitled to produce more oxidative stress."
- The authors thank the Reviewer. The sentence in line 641 has been modified in “. Exposure to pollutants triggers oxidative stress and inflammatory signaling which increase as the surface area of ultrafine particles rises.”
“Table 2: None of the articles listed have an included citation.”
- The authors thank the Reviewer. All the references were provided.
“Line 731: "exosome" seems to be "exposome".”
- The authors thank the Reviewer. Issues highlighted in line 731 have been fixed in the text.
“Figure 2 could be improved as it looks to be of low resolution.”
- The authors thank the Reviewer. The figure resolution was improved and ppt file was provided.
Reviewer 2 Report
Comments and Suggestions for Authors
This is an interesting minireview focused on Exposome from endogenous and exogenous factors in non-communicable chronic diseases. The recent publications were digested and re-organized to provide a picture for the general audience. However, this is a review type article, not simply a “literature digestion”, and the presentation needs significant improvements. Listed below are specific comments:
1. Table 1 should be correlated with “Endogenous factors” in the text with subheadings as “Genetics and epigenetics”, “Gut microbiota”, “Aging” and “Diabetes” and references follow the major points, rather than presented as a “literature study”. Factors first, followed by description, and citations last. When article number is used, the title and article type are quite unnecessary.
2. Table 2 should be correlated with “Exogenous factors” in the text with subheadings as “Diet” (Endocrine-disrupting chemicals, Vitamins, AGEs, 1,2-dicarbonyl compounds, Fructose etc.); and “Pollutants” (PM, Ozone, Metals, Pesticides, Indoor air pollution etc.). The major points first, followed by description and article citations last. When article number is used, the title and article type are quite unnecessary.
3. Others, for example:
a. Line 35: “This review of reviews investigates the exposome's role” could be “This paper reviews investigations on the exposome's role”
b. Line 167: “capability to answered to life stressors” could be “capability to respond to life stressors”
c. Line 294: “96 childhood exposures” could be “96% of childhood exposures”
d. Line 655: “Environmental pollutants known as EDCs” could be “Environmental endocrine-disrupting chemicals known as EDCs”.
e. Line 808: “A One Health approach” is unclear.
Comments on the Quality of English Languagegenerally OK
Author Response
Dear Editor of Diseases
First, my coauthors and I would like to thank You sincerely for this opportunity of cooperation. We profoundly thank the reviewers for the comments and useful suggestions aimed at improving the paper. We thank You for your constructive critique and we hope the review process has led to an improved manuscript. If additional changes are warranted, we will make them.
We hope that this revised version of our manuscript may now be found suitable for publication.
This is a point-by-point list of changes made in the paper:
Reviewer 2
“This is an interesting minireview focused on Exposome from endogenous and exogenous factors in non-communicable chronic diseases. The recent publications were digested and re-organized to provide a picture for the general audience. However, this is a review type article, not simply a “literature digestion”, and the presentation needs significant improvements. Listed below are specific comments:
- Table 1 should be correlated with “Endogenous factors” in the text with subheadings as “Genetics and epigenetics”, “Gut microbiota”, “Aging” and “Diabetes” and references follow the major points, rather than presented as a “literature study”. Factors first, followed by description, and citations last. When article number is used, the title and article type are quite unnecessary.”
- The authors thank the Reviewer. Table 1 was revised according to his/her comment.
“2. Table 2 should be correlated with “Exogenous factors” in the text with subheadings as “Diet” (Endocrine-disrupting chemicals, Vitamins, AGEs, 1,2-dicarbonyl compounds, Fructose etc.); and “Pollutants” (PM, Ozone, Metals, Pesticides, Indoor air pollution etc.). The major points first, followed by description and article citations last. When article number is used, the title and article type are quite unnecessary.”
- The authors thank the Reviewer. Table 2 was revised according to his/her comment.
“3. Others, for example:
- Line 35: “This review of reviews investigates the exposome's role” could be “This paper reviews investigations on the exposome's role””
- The authors thank the Reviewer. The sentence in line 35 has been modified in “This study investigates […]”.
“b. Line 167: “capability to answered to life stressors” could be “capability to respond to life stressors””
- The authors thank the Reviewer. The sentence in line 167 was revised according to his/her comment.
“c. Line 294: “96 childhood exposures” could be “96% of childhood exposures””
- The authors thank the Reviewer. After carefully rechecking the study (DOI 10.1289/EHP5975), it is confirmed that 96 is a number and not a percentage.
“d. Line 655: “Environmental pollutants known as EDCs” could be “Environmental endocrine-disrupting chemicals known as EDCs”.”
- The authors thank the Reviewer. The sentence in line 655 was revised according to his/her comment.
“e. Line 808: “A One Health approach” is unclear.”
- The authors thank the Reviewer. The sentence in line 808 was revised as follows: “A One Health approach, which recognizes the interconnectedness of human, animal, and environmental health, is essential to addressing these complex challenges. This integrated approach considers how the health of humans is closely linked to the health of animals and the environment in which they live.”
Sincerely
Rossella Cianci
Round 2
Reviewer 2 Report
Comments and Suggestions for Authors
The authors have carefully addressed the questions from the reviewer and made corresponding corrections in details, the manuscript has been improved to tell a good story. This reviewer does not have additional questions.